# The language of marketing hyperbole and consumer perception–The case of Glasgow

Sean MacNiven[1,2]*, J. John Lennon[2], Julie Roberts[2], Maxime MacNiven[1,3]

1 SAP SE, Walldorf, Germany, 2 Glasgow Caledonian University, Glasgow School for Business & Society, Glasgow, Scotland, United Kingdom, 3 SRH Heidelberg, Heidelberg, Germany

☯ These authors contributed equally to this work.

* sean.macniven@sap.com

**Data Availability Statement:** The data underlying the results presented in the study are available from https://osf.io/amscb/.

## Abstract

The aim of the study was to explore the potential of a timed word association task to generate detailed insights into the perception of Glasgow city and its people which could inform destination and brand marketing. Destination marketers have a challenge to convey the tourist destination image to attract and satisfy the expectations of its visitors. Yet destination perceptions are often the result of multiple tourist visitor experiences at a location, neglecting the voice of the resident. The extent to which word associations varied by participants' relationship to Glasgow was identified in terms of Aaker's brand personality scale, an extension of personality research on brands and destinations. Surveying of 1,219 UK participants generated a total of 5,993 terms (city; 1,144 unique) and 5,034 terms (people; 944 unique). The value of capturing the perceptions of a destination by its residents is identified. The results showed that the city of Glasgow was often described as cold and busy, while the people were primarily described as friendly and funny. Evidence was found in support of dual-processing theory suggesting word associations based on lived experiences of a city may be generated later (in terms of the order in which the terms were generated) in a word association task, while common linguistic associations (e.g. synonyms, antonyms, hierarchies etc.) tend to be generated earlier in the task. As hypothesised, analyses revealed a significant relationship between several of the Aaker-dimensions of brand personality, and the consumers' relationship to Glasgow, extending marketing research with an empirical approach to identifying differences in the perceived personality of a destination. The study offers a practical, fast, and replicable method for destination marketers to study consumer perception at scale, which is currently not widely utilised in this field. In particular, the use of semantic distance and word embeddings provides a readily available approach to automatically categorise content derived from word associations studies, or indeed, any text-based content. In contrast, financial investment in non-validated branding and destination marketing campaigns appear to be increasingly problematic. Advances were made in testing an approach to interpreting word associations through the lens of linguistic and situated simulation (LASS) theory to provide deeper analysis to both categorise and interpret consumers' perception. Traditional approaches to tourism marketing and destination branding rarely provide such a level of analytical appraisal. The analysis presented in this paper challenges the orthodoxy and validity of investment in brand and destination marketing at a city level

**Funding:** The author(s) received no specific funding for this work.

**Competing interests:** The authors have declared that no competing interests exist.

and the potential for word association tasks to be used as a valuable alternative method to create more effective destination marketing and branding.

## 1. Introduction

The rise of global urban tourism looked inexorable until March 2020 and the advent of the global pandemic. The closure of international borders and the reduction in air travel to control infection changed this growth trajectory. Urban centres emptied of workers, leisure tourists, business visitors and students. As the pandemic retreats, cities now face further challenges through inflation and energy crises, and an ongoing war in Ukraine and its impact on business and the global economy [1]. For Glasgow, this has meant a reduction in visitation and tourism revenues [2], it has lost ground in terms of performance, ranking 118 out of 362 UK cities on competitiveness [3]. The effectiveness of city branding and destination perceptions, is particularly relevant to support the growth of a city. Perceptions of urban destinations are promoted through branding and destination marketing to encourage visitation and inward investment. In this paper, Glasgow is used as a case city, to test the value of a word association task to identify destination perception in the post-COVID-19 period.

Cities generate economic growth and are critical to national performance, urban tourism has been a further facet in building competitive advantage for progressive cities [4]. The combination of transportation and particularly air connectivity with other urban centres has ensured a regular supply of leisure and business visitors. As connectivity has become more cost effective, following the rise of budget air travel, the appeal of cities has continued to grow (Page and Connell, 2020). They combine significant leisure appeal including offering experiences in heritage, festivals and events, retail, nightlife with a range of cultural attractions. For business visitors, urban destinations offer; business locations, meetings, and conference centres as well as a range of potential markets and consumers concentrated in one location. Meeting Incentives Conferences and Events (MICE) tourism has become synonymous with cities as they constitute the primary locations of such business. A virtuous circle of leisure/business demand and transport connectivity has been the key to cities becoming major drivers of tourism.

The urban data produced by the World Travel and Tourism Council [5] confidently reported that global urban destinations were out performing all other locations. For those involved in marketing cities, the relationship between; *Positioning* (how the city should be perceived); *Image* (how people perceive the city); *Branding* (how the cities desired or aspirational image is expressed) and *Promotion* (how the city's branding is communicated) are considered critical [6, 7]. For most urban destinations the branding is a critical and iterative association that matures whilst embedding perceptions through various communications channels [8]. In Glasgow, the global COVID-19 pandemic changed this growth dynamic reducing urban visitation (Moffat Centre, 2021; 2022, 2023). In the UK accommodation occupancy and rates achieved were much greater in non-urban locations during 2021 and 2022 as leisure demand migrated away from cities in the immediate post-COVID-19 environment. Business and conference demand was also diluted by the continued use of online channels and the lower footfall within cities following work from home practice becoming embedded in general employment [3].

Defining and building a brand requires a clear understanding of perception in terms of the destination and its people who live there, across various demographics and relationships to the

destination, yet such comparisons are rare, and tourists remain the subject of most tourism research, with local perceptions poorly represented [9]. Digital trace data generated when interacting with online media, such as Twitter, Facebook, Trip Advisor, Booking.com and similar are a rich source of such perception, however, they are also archival and observational [10–12], and not the results of a controlled process. One promising means of both generating new insights and testing existing theories, can be found through studies of free word associations, the space those words inhabit, and the company they keep (i.e. co-occurrence and collocation) [13, 14].

Perceptions of Glasgow, and the ways in which those perceptions vary by way of the strength of the relationship to the city and its people, has to date, never been explored at scale, and represents a critical opportunity for destination marketers, especially given the novel challenges emerging from a post-COVID-19 world.

## 2. Literature review

### 2.1 Urban tourism

In 2019, Scotland evidenced its greatest rise in international and domestic tourism [15], invariably such growth can positively or negatively impact host, resident / native communities. Host communities' acceptance of tourism development can influence destination performance [16]. Sharpley [17] identified notable tourism-specific factors, such as; type of tourism, density of tourists, or tourism dependency as elements impacting host perceptions. San Martín et al. [18] considered cities, regions or countries to be 'brands' in the eyes of their populations. Marketers have a challenge to convey the tourist destination image to attract and satisfy the expectations of its visitors, yet destination perceptions are often the result of multiple tourist visitor experiences at a location [19]. However, a focus purely on tourists and visitors omits the opportunity to obtain impressions from locals, whose identities define and are defined at least partially by the cities they inhabit [20]. Indeed, the growing trend among travellers to seek "...authentic, experientially oriented opportunities with more meaningful interactions with locals" [21] would highlight significance. Thus, obtaining insights into how locals and visitors perceive urban destinations would appear to be especially timely in understanding urban appeal [20].

Urban tourism has been long associated with place branding and marketing [7] and has been the subject of research ranging from its use in classic heritage cities such as Paris [22] to later emergent urban destinations such as Dubai [23]. In the case of Scotland, Glasgow's investment in place branding and resident and visitor perceptions merits review in the post-COVID-19 environment. Glasgow has been cited as a positive example of economic revival through urban tourism development [24, 25]. This city, in redressing negative perceptions associated with post-industrial deprivation and manufacturing decline; reinvented itself as an urban tourism destination. The combination of visitor attraction development, staging of landmark events (including the 1988 Garden Festival, the 1990 European City of Culture to the 1996 Year of Architecture and Design and the 2014 Commonwealth Games) served to build awareness and drive visitation. The development of leisure tourism was balanced by investment in conference and meeting infrastructure, notably the Scottish Exhibition and Conference Centre (1986); which was followed by development of the Scottish Event Campus, following construction of the Armadillo meeting facility (1995) and the Hydro (2013) Performance and Conference Arena. The city also saw investment in tourism infrastructure that included major refurbishments of Kelvingrove Museum and Gallery (2006), development of the Riverside Museum (2011) and the Burrell Collection refurbishment (2022). During the same period significant investment was occurring in destination branding. Frequently, such investment has failed to recognise the complex range of motivations influencing choice [26].

Human motivation is extremely complex and decision making is multi-causational. Influences ranging from accessibility to visual appeal, price and security, wider media coverage and historical perceptions are combined with numerous other influences questioning the veracity of place marketing.

## 2.2 Destination branding

Place marketing has largely been informed using qualitative analysis [27] although the work of Prebensen [28], is a useful exception. In the case of destination branding, the *Glasgow's Miles Better* place marketing campaign was associated with then Lord Provost; Michael Kelly [29] and utilized the Roger Hargreaves' Mr. Happy cartoon [30] in what has been seen as a cited example of successful urban marketing [24, 25].

This was followed in 2004 by the *Glasgow*: *Scotland with Style* campaign led by Greater Glasgow Tourist Board (GGTB). It was seen as a successful urban marketing body with a track record in sourcing, attracting, and hosting sporting, cultural and corporate conferences and events whilst promoting Glasgow as a desirable destination for leisure and business tourism.

The place marketing of Glasgow matured from the original slogan led, municipally inspired campaign to more aspirational approaches to city branding. The *Glasgow; Scotland with Style* campaign communicated a generic and flexible set of messaging about the city featuring; leisure, business, retail, dining, conferences and exhibition through sophisticated imagery and narratives.

This approach was based on an augmented Brand Wheel of place rationale and process (Fig 1). Such an approach highlights the current and aspirational perceptions of place using key descriptive words and phrases to guide destination marketing [31]. The Brand Wheel was used in a range of destinations at national level and regional level.

The focus on the attributable value of style is important [32] with its symbolic and positive association with Glasgow architect; Charles Rennie Mackintosh. This was the result of content analysis of Glasgow travel reviews undertaken by Glasgow City Marketing Bureau (GMB) [25] and formed the basis of place marketing. As this was produced by brand consultants, who maintained commercial confidentiality in respect of derivation of content and descriptors making validation of the Brand Wheel impossible. The perceived success of the brand meant it was retained until 2016, featuring in the 2016 Tourism Strategy authored by: Glasgow City Council, VisitScotland, Glasgow Chamber of Commerce, Scottish Enterprise Glasgow and Glasgow City Marketing Bureau. It concluded:

> *"Our vision for Glasgow 2016 is of a leading destination in key markets offering a unique, dynamic and authentic experience through the quality of place, product and service differentiated through the strength of the brand, Glasgow*: *Scotland with style."* [33]

The claims for return on investment from the *Style* branding are difficult to validate, yet in 2007; GMB claimed £62 million in local economic benefit attributed to the brand and marketing campaign. However, this was little more than attribution of all tourism expenditure over the period 2005–7 whilst discounting any other factors that could motivate visitation. Any attempt to delineate and quantify the range of factors and influences causing consumers to visit Glasgow would be problematic. Such place branding exists to market city destinations whilst providing a unifying brand for stakeholders and business intended to positively influence consumer perceptions [7].

The *Style* branding was replaced in 2013 with *People make Glasgow* to broaden city coverage beyond the style and design orientation which, whilst favouring retail and design was proving

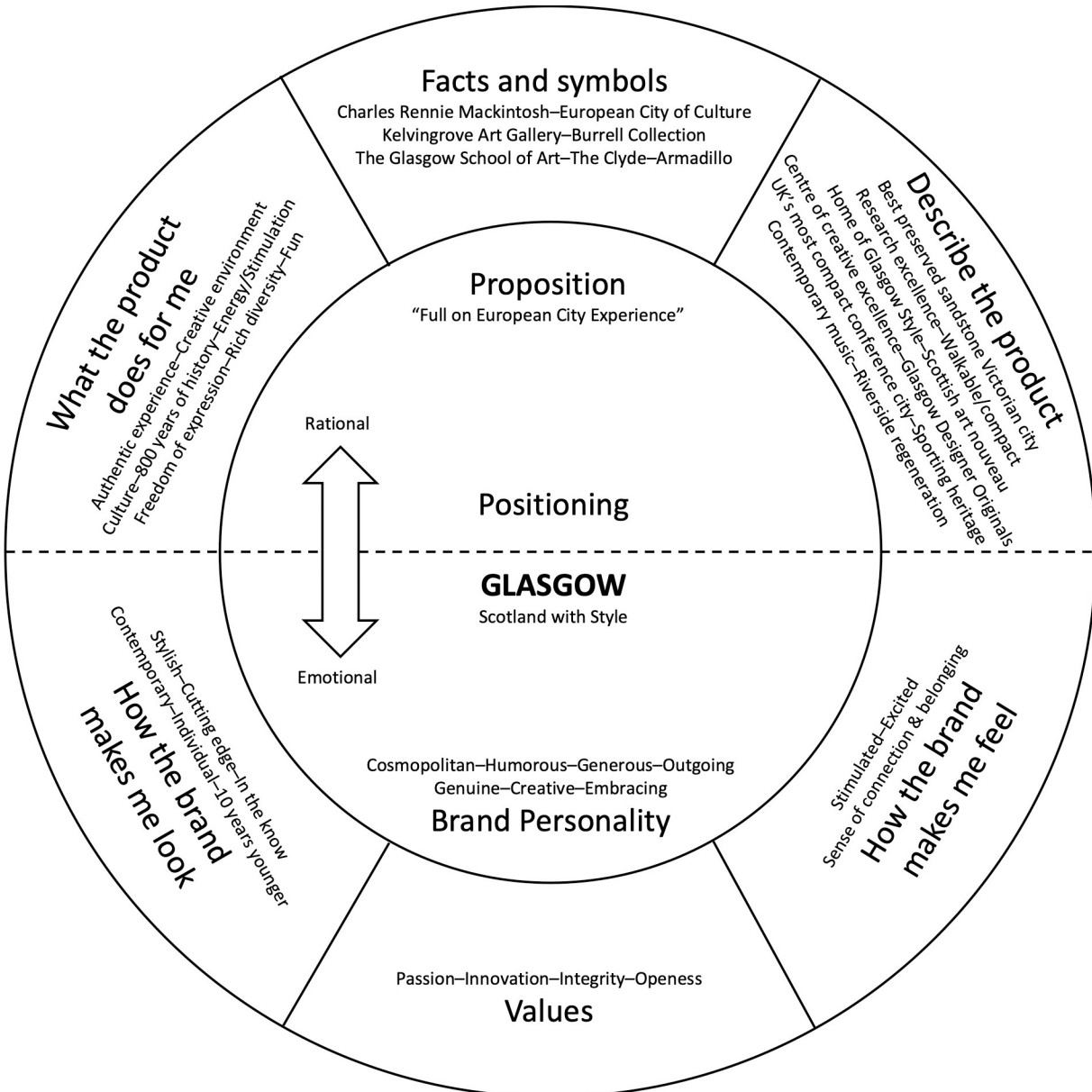

**Fig 1. Brand Wheel (authors' reproduction of the Glasgow visitor marketing Bureau 2004).**

less attractive to broader business and emergent financial sectors. This brand which has been in use continuously since 2013, provides a broad and flexible premise; *'People make. . .'*. This can be combined with a range of sectoral meanings; *People make Glasgow Welcoming; People make Glasgow Real; People make Glasgow Creative.*

The premise being used to drive and influence perceptions, frequently combined with imagery, this drives a further positive association for the city. The value of destination perception is tested in this paper wherein word association on sample populations are used to consider effectiveness of such marketing investment. Traditional approaches to tourism marketing and destination branding rarely provide such a level of analytical appraisal [see for example 34].

## 2.3 Word association

In tourism and travel research, word associations have been used to determine the perception of distant destinations [28, 35, 36], where such associations are understood to be shaped by the experiences, memory, and imagination specific to each individual [37]. Indeed there is evidence that perceptions vary substantially across native/indigenous populations versus tourists [27, 38, 39], not to mention how perception may vary among those who may only have heard about a destination and never visited. It follows that controlling for the conceptual variability among word association task participants regarding a destination cue may be of particular importance and relevance to tourism and travel research [40].

One source of conceptual variability may be found in the process of generating associations themselves. Paivio's pioneering work on Dual Code Theory (DCT) hypothesised that both language and imagery work together strengthen associations and learning [41, 42], while Barsalou, building upon DCT, found strong support for the role of simulation when generating associations, leading to the development of the linguistic and situated simulation (LASS) theory [43, 44]. Linguistic associations may include properties conceptually common to a city such as *large* or *busy* [45], as well as properties that capture unique personal experiences, for example *the pie-shop under the bridge* [46]. This was further supported by the results of a Functional Magnetic Resonance Imaging (fMRI) study to chart the response types in the brain, finding that linguistic associations precede the richer, more individually situated simulations, e.g. *pie shop* [47]. According to LASS, word associations utilise "alternative sources of information: the linguistic form system and the situated simulation system" [46]. Consequently, LASS predicts that the order of delivery of words in a timed word association task varies by way of the generating mechanism, with linguistic properties proceeding simulations. The present study employs LASS as a theoretical lens to both categorise and interpret the properties generated and provide an additional opportunity to test predictions made by the theory. To date, no study of destination marketing could be found exploring the role of generation order of terms in word associations studies, nor differences in personal associations versus culturally embedded interpretations of the city concept.

## 2.4 Considering destinations as people

The Aaker brand personality scale is a tool used in marketing and branding to assess and measure the perceived personality traits of a brand. It comprises of five *brand personality* dimensions which are; *Sincerity*, *Excitement*, *Competence*, *Sophistication*, and *Ruggedness* with each further extended by additional adjectives, resulting in a set of 42 descriptors [48]. Based on the premise that a location too is both a brand and can be described in terms of a set of human characteristics, several attempts have been made to apply the dimensions to destination branding. A study of the perception of two Spanish destinations among UK tourists confirmed the validity of all five of the Aaker dimensions [27, 35, 49], while a 2017 study of destination in terms of *personality fit* concluded that the match between tourists and the perceived personality of a destination serves to "reinforce the importance of strong and distinct destination personalities" [50]. Although some attempts have been made to automatically infer the Aaker dimensions from social media content, for example considering personality and brand in Thailand [51], city branding responses to COVID-19 [52] and the traits of world heritage sites [53], automated destination personality assessment represents a promising opportunity that has only begun to be explored.

## 2.5 Self versus other: Residents and tourists

Although tourism research has traditionally focused on the perceptions of tourists, with the perspectives of other stakeholders, such as residents "underrepresented in the existing

literature of place" [9], the idea that perception may vary by relationship to a destination has been gaining traction in the literature. A study of the emerging tourist destination of Molise, Italy found substantial differences among residents and tourists in their perception of and propensity to recommend the location [54]. Similarly, while a study into resident versus international visitor perception of Thailand found both groups shared opinions regarding sightseeing, friendliness and food, visitors placed a higher importance on nightlife and entertainment than residents [55]. Indeed, a 2005 study of visitors to Florida found substantial differences not only in perception of residents, it also found differences across visitors in terms of U.S. domestic versus international [56]. Despite these findings, degrees of incongruence of opinion across residents and visitors may not apply universally, with a study of visitor and resident perception of Waiheke Island, New Zealand finding a high degree of congruency regarding perception of the island among visitors and residents [57]. While it remains to be explored what the degree of (in) congruence might mean for a destination, the extent to which this might apply to the City and People of Glasgow represents an interesting research question, especially for destination marketers.

## 2.6 Hypotheses

A key premise of this paper is the idea that "you shall know a word by the company it keeps" [58]. The company a word keeps is defined by its semantic distance and location relative to other words as derived from large text corpuses, also known as word embeddings. For example, the word bee will often be found close to words such as hive, and sting, and less frequently close to the word metallurgy. Word embeddings allow for the meaning of words to be represented mathematically, and to facilitate computational approaches. In section 2.3 the potential to interpret word associations through the lens of LASS was considered, while in section 2.4, the role of brand as personality, using the Aaker model was considered. From both the theoretical perspective of LASS, and practical perspective of destination marketing, the following hypotheses will be tested.

H1: Word associations written later are predicted to be more distant from the city and each other.

a. Last words are more distant to [Scotland], [Glasgow], [city] and [town] than first words across all respondents.

b. Word associations written later are predicted to be more distant from each another (greater variance).

Word associations generated later in the word association task by the by the respondent are more likely to be situated simulations (SS) than language associations (LA). Specifically, SS are predicted to be more distant from the core concept of city (extended to tests of Scotland and Glasgow as sub-categories) than LA, and to have higher variability (i.e., terms generated later will be more unique from one another in terms of average semantic distances than the average distances calculated for LA).

As discussed in Section 2.5, there is considerable evidence to suggest that how residents and visitors perceive a destination may vary substantially. Where the Aaker scale seeks to provide a universal instrument to capture the essence of a destination, the present study hypothesises that a destination may have two or even several personalities, when segmenting by relationship to the destination. It follows, therefore, that:

H2: Self-perception (*residents*) of Glasgow city and its people, and other-perception (*visitors* and *never-visited*) will differ significantly across coded brand dimensions. Terms generated by respondents will vary by relationship to Glasgow in terms of their Aaker distribution.

Following guidance around the preregistration of confirmatory studies regarding the potential value of including subsequent exploratory research or post-hoc analysis (e.g. https://plos.org/open-science/preregistration) in addition to these pre-registered hypotheses, relationships and categories beyond the Aaker dimensions were explored, including differences across demographics as both original insights and opportunities for future research.

## 3. Methods

The following section provides an overview of our methods. In support of both validation and replication by others, the full data and notebooks are available here: https://osf.io/amscb/. Many additional charts and tests are available for other researchers to explore, but not included here as they were not immediately relevant to the paper and its hypotheses. The survey can be found in S1 Appendix. The participants' tasks were timed at 25 seconds for "City of Glasgow" and a second task and timer for the "People of Glasgow", to include sufficient time for the generation of both word associations and situated simulations in accordance with prior research [46]. All demographic data were obtained directly from the *Prolific online survey tool*, with the exception of education. An additional question was added to determine the relationship of the participant to Glasgow (i.e. whether they had ever lived in or visited the city), and for those who had been to the city, several questions around tourist highlights followed the word association questions (see S1 Appendix for the complete survey).

### 3.1 Ethics

An ethics application was approved by the Glasgow School *for* Business and Society (GSBS) ethics committee to ensure all due ethical considerations were taken in the study design, as the research study involved human participants. Participants had to be 18 years or older, and so no minors were involved in the study. The following written explanation of the project was provided through the Prolific service advertisement at the start of the survey, explaining the aim and benefits of the study, as well as the expected remuneration:

> *The aim of the study is to explore the perception of Scottish cities through word associations in order to help local governments better understand urban locations. If you participate, you will be asked to write as many words as you can think of for two aspects of one Scottish city, in 25 seconds. The association task is then followed by a short series of questions to understand your relationship to the city, and some basic demographics. All in all, the study should take no longer than a few minutes to complete (7–9 questions in total).*

Users were asked to provide their Prolific ID in order to obtain demographic information they had previously provided to the service. Any personal information linked to this ID was not stored with the survey data and was completely inaccessible to the research team) and no additional personally identifiable information was requested or obtained by the research team, rendering the survey data anonymous. A box to check for participants to provide consent prior to participation was also provided.

### 3.2 Participants

A timed, free word association task was administered to a group of (N = 1392) respondents recruited via the panel service Prolific from April 3, 2022 to May 2, 2022. The minimum sample size required for the UK was calculated for a 5% margin with a z-score of 1.96 at 370. Participants were recruited past this minimum in order to increase accuracy. The sample was based on self-selection and all complete and valid survey responses were included in the study.

**Table 1. Demographics.**

| Relationship to Glasgow | Never visited | | Visited but never lived there | | Lived or living there | | Full sample | |
|---|---|---|---|---|---|---|---|---|
| | n | % | n | % | n | % | n | % |
| Sex | | | | | | | | |
| Female | 304 | 58.0% | 246 | 58.7% | 179 | 64.9% | 729 | 59.8% |
| Male | 217 | 41.4% | 171 | 41.1% | 98 | 35.1% | 486 | 39.9% |
| Prefer not to say | 3 | 0.6% | 1 | 0.2% | 0 | 0% | 4 | 0.3% |
| Employment status | | | | | | | | |
| full time | 254 | 48.5% | 231 | 55.1% | 151 | 54.7% | 636 | 52.2% |
| part time | 90 | 17.2% | 82 | 19.6% | 63 | 22.8% | 235 | 19.3% |
| student | 83 | 15.8% | 38 | 9.1% | 34 | 12.3% | 155 | 12.7% |
| not employed | 61 | 11.6% | 29 | 6.9% | 21 | 7.6% | 111 | 9.1% |
| retired | 19 | 3.6% | 24 | 5.7% | 5 | 1.8% | 48 | 3.9% |
| disabled | 17 | 3.3% | 15 | 3.6% | 2 | 0.8% | 34 | 2.8% |
| Country | | | | | | | | |
| England | 458 | 87.4% | 308 | 73.4% | 82 | 29.7% | 848 | 69.6% |
| Scotland (Glasgow) | 1 | 0.2% | 21 | 5.0% | 128 | 46.4% | 150 | 12.3% |
| Scotland (Not Glasgow) | 7 | 1.3% | 48 | 11.5% | 43 | 15.6% | 98 | 8.0% |
| Wales | 19 | 3.6% | 12 | 2.9% | 4 | 1.4% | 35 | 2.9% |
| Northern Ireland | 11 | 2.1% | 12 | 2.9% | 3 | 1.1% | 26 | 2.1% |
| No Information | 28 | 5.4% | 18 | 4.3% | 16 | 5.8% | 62 | 5.1% |
| **TOTAL** | **524** | **43.0%** | **419** | **34.4%** | **276** | **22.6%** | **1219** | **100%** |

N = 1219. Participants were on average 36.3 years old (SD = 12.6), all were resident in the UK spoke English. 5993 terms generated, and averaging 5.92 words for Glasgow city, and 4.13 for people.

From those participants, 135 were dropped due to missing demographics in the prolific database. Another 39 participants had not completed the survey and were therefore excluded from analysis.

After cleaning the dataset, 1219 participants were eligible for analysis. Of those participants, N = 277 were residents of Glasgow or had lived there at some point in their life, N = 418 had visited the city but never lived there, and N = 524 had never visited Glasgow (see Section 4, Table 1 for details).

## 3.3 Natural Language Processing

Natural Language Processing (NLP) is a common approach to the automated analysis of large bodies of text, that has been frequently applied to the categorisation of words in word association tasks and is a frequently used method for exploring human cognition and perception [59–62], such as creativity [63] stereotypes [64] and social biases [14]. Most immediately relevant to the present study, it has also been used to explore tourist perception in blogs [65] and general tourism data mining [66].

To clean the raw word associations, and reduce complexity, several layers of natural language processing were applied. After removing special characters and redundant spaces, all typos were identified by collecting all the words that were not identified by an English dictionary library in python, the widely used programming language, and corrected manually. Furthermore, all plural words were converted to their singular form. As there was no standardized format to identify bigrams or trigrams, such as hard-working or down-to-earth, all possible combinations of bigrams and trigrams were collected programmatically and sorted by

frequency. The n-grams that were most likely to be actual n-grams were then selected by the researchers.

## 3.4 Distributional semantic networks and LASS

Distributional semantic networks (DSNs) utilize the co-location of words to generate a distributional topology of concepts as they naturally occur in human language generation, including the study of situated versus linguistic processing [67]. Associative strength is conceptualised in terms of the frequency of associations, for example if 45 out of 100 of people generate the word STING in relation to the cue word BEE, the associative strength would be 0.45. Such strong links are likely to be generated by linguistic mechanisms, whereas particularly weak intra-lexical links are likely to result in idiosyncratic responses [68]. It follows then, that in terms of DSNs, words that are frequently collocated should have higher associative strength and be semantically closer to one another than those generated by the more idiosyncratic simulation system [69].

In order to test the potential for automatic identification of situated simulations versus language associations, simulations are hypothesized as being the result of individual experiences and, therefore, less generic than language associations. This idea of uniqueness translates well to the idea of semantic distance whereby similar words (in terms of the frequency of their co-occurrence) will be semantically close, whereas less commonly co-occurring terms will have greater semantic distance, a notion that has been successfully applied to research around creativity (Beaty et al., 2022). This interpretation is strongly supported by the coding scheme from Santos et al. [46], which describes language associations in terms of word types expected to have a high co-occurrence (antonyms, synonyms, hierarchies, etc.). In this way, situated simulations are operationalised as having greater semantic distance from the core concept of interest (here, Glasgow) than language associations. To minimise bias through framing effects or anchoring, the survey began immediately with the word association tasks and follow-up questions were then presented later in the survey.

To calculate semantic distance, the cosine distance between two given word vectors was calculated using the well-known GloVe-algorithm [59–62], trained on the "fasttext-wiki-news-subwords-300" text corpus. In a similar fashion, the variance between all first and all last words, respectively, were calculated.

## 3.5. Hypothesis testing

To test H1a, four t-tests for dependent samples were used to compare the distance of first and last words to each of the terms *Glasgow*, *Scotland*, *town* and *city*. As each participant has two instances of first and last words from the two successive association tasks, the distance values were averaged over the two tasks before running the t-tests.

H1b stated that "Word associations written later are predicted to be more distant from each another". To test this hypothesis, the variance of associations was calculated within the group of all unique first, and all unique last words, respectively. This was done by calculating the mean semantic distance of each word to every other word within that group, analogous to the methodology of Olsen [70]. The difference in the average distances between unique first words and the average distances between last words were then tested using an independent sample t-test.

To test H2, a logistic regression was modelled on the data. The dependent categorical variable was relationship to Glasgow, with resident (1) and never visited (0), dropping the visitor's category to achieve two groups. Two separate models were run, one including only Aaker-dimensions and the number of words generated (to control for the fact that visitors may have

used more words on average, increasing the probability of a positive significant effect). The second model used the same structure, but additionally controlled for sex, age, highest achieved education, and employment.

### 3.6 Coding and sorting of words into Aaker-Dimensions

Traditionally, Aaker-Dimensions are measured by giving participants the opportunity to rate 5–11 adjectives per dimension on a 5-point agreement scale [48]. The aim of this analysis was to see if we can generate Aaker-scores using the word associations. Specifically, we wanted to see whether semantic similarity algorithms could be used to meaningfully assign adjectives and nouns into each Aaker dimension. For this analysis, slang words were removed due to high ambiguity.

Negative words were also removed as the classical Aaker dimensions only contain neutral and positive adjectives. With the remaining set of words, the average semantic distance to all adjectives for each Aaker-dimension was calculated. Each word was assigned to a dimension with the lowest semantic distance if the semantic distance was below a certain threshold and hence sufficiently close to that dimension. The threshold was chosen by the authors, striking a balance between the inclusion of relevant words and the exclusion of words unbefitting the assigned Aaker-dimension. This left an array of words with a unique Aaker-dimension, and a list of words with no dimension.

Table 2 shows the top 15 words for each dimension. Each participant could have one, several or none of the Aaker dimensions, depending on how many categories are represented in their word-association. For example, a participant with the associations "busy" (Competence), "ice-cream" (None), "rough" (Ruggedness) and "tough" (Ruggedness), would be assigned a "1" for the Aaker-Dimensions of Competence and Ruggedness, and a "0" for Sincerity, Excitement and Sophistication.

### 3.7 Exploratory factor analysis

In addition to the pre-registered hypotheses, additional ad hoc analyses were conducted to explore what other terms outside of the Aaker dimensions may have emerged in descriptions

**Table 2. Word frequencies by Aaker dimensions.**

| Competence | WC | Excitement | WC | Ruggedness | WC | Sincerity | WC | Sophistication | WC |
|---|---|---|---|---|---|---|---|---|---|
| busy | 141 | football | 143 | cold | 169 | friendly | 659 | accent | 265 |
| people | 122 | fun | 133 | rough | 157 | funny | 231 | beautiful | 31 |
| university | 71 | city | 119 | tough | 82 | loud | 230 | pretty | 25 |
| drug | 69 | culture | 106 | kilt | 77 | kind | 116 | white | 25 |
| hard | 59 | art | 64 | north | 36 | nice | 91 | working-class | 23 |
| building | 58 | urban | 61 | tartan | 31 | helpful | 89 | dark | 21 |
| proud | 58 | big | 61 | river | 29 | happy | 81 | tenement | 20 |
| strong | 57 | music | 53 | rain | 29 | welcoming | 79 | rowdy | 18 |
| large | 42 | shopping | 49 | bagpipe | 26 | old | 75 | lovely | 16 |
| home | 41 | green | 42 | northern | 23 | haggis | 68 | blue | 15 |
| industrial | 26 | museum | 41 | wet | 21 | warm | 56 | cultured | 15 |
| party | 25 | shop | 39 | rainy | 20 | drinking | 45 | statue | 14 |
| hard-working | 24 | vibrant | 37 | hardy | 17 | down-to-earth | 43 | ginger | 14 |
| outgoing | 23 | diverse | 33 | scot | 16 | alcohol | 41 | mackintosh | 13 |
| train | 21 | drink | 33 | snow | 15 | beer | 41 | brash | 13 |

of Glasgow and its people, and how might these be used to extend of enhance the Aaker model.

To see if the Aaker personality dimensions were utilised by the sample in their timed responses, an exploratory factor analysis with varimax rotation was run separately for both city and people as a method for investigating the co-occurrence of words. This was achieved by creating a binary column for each term of the word association task (for example, "park"), to indicate whether a participant had, or had not used the term. Then, to ensure that the words used had sufficient weight and relevance, words with fewer than ten occurrences were removed. Words that tend to appear together in the same response, were placed into a common category.

## 4. Results

### 4.1 Participants

Table 1.

### 4.2 Word frequencies

**4.2.1 Words by cue (city and people).** Fig 2. provides the main word associations made with Glasgow. Positive descriptors such as friendly, funny, and fun, as well as adjectives referring to the rustic features of Glasgow such as cold, rough and busy, are among the most common associations. Accents, football and culture are nouns frequently mentioned in the context of Glasgow. While the city itself was often described as cold and busy, the people were primarily described as friendly and funny. In the following figures (Figs 3 and 4), these overall trends are displayed separately for different demographic groups.

**4.2.2 Words by age.** Descriptions for the city of Glasgow differs especially for people above the age of 55, for whom culture and football are much more common associations than

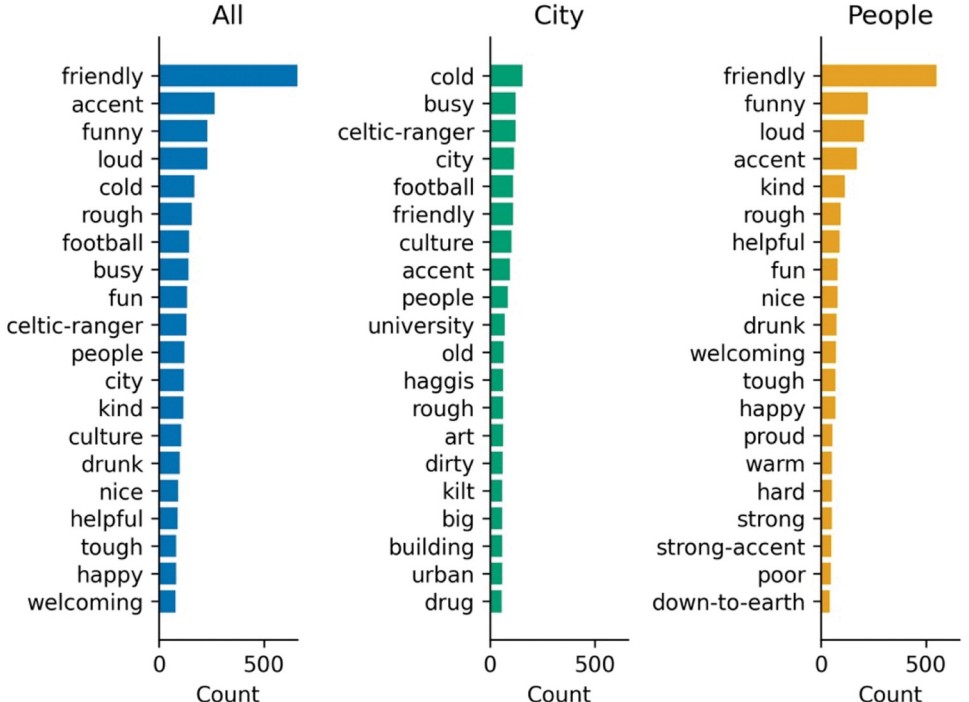

**Fig 2. Overall word frequencies by cue (city and people).**

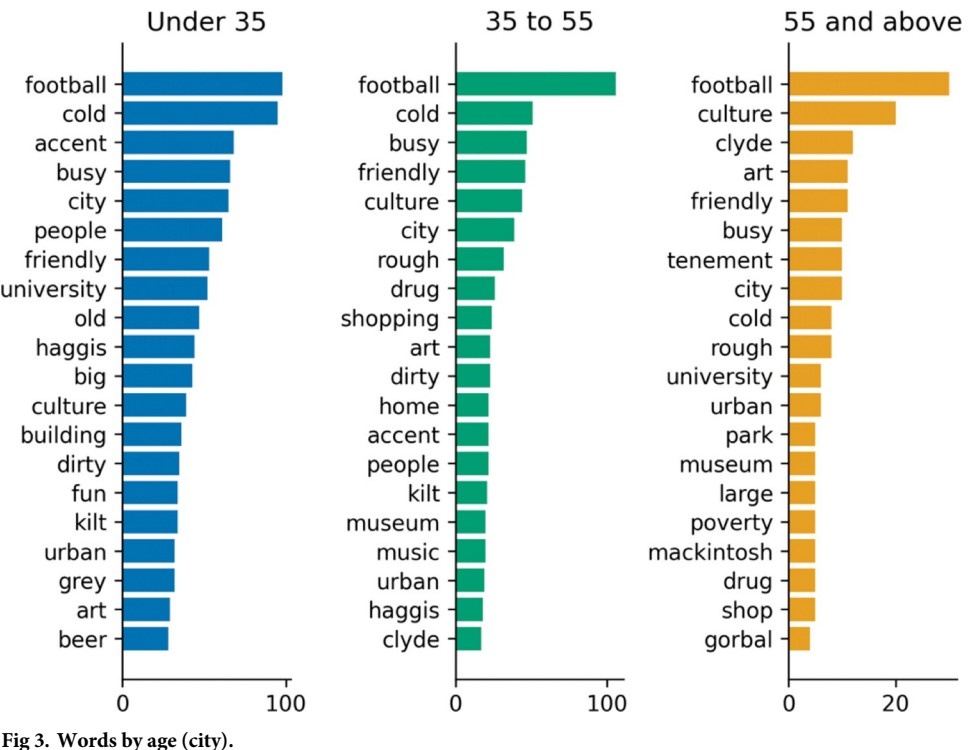

**Fig 3. Words by age (city).**

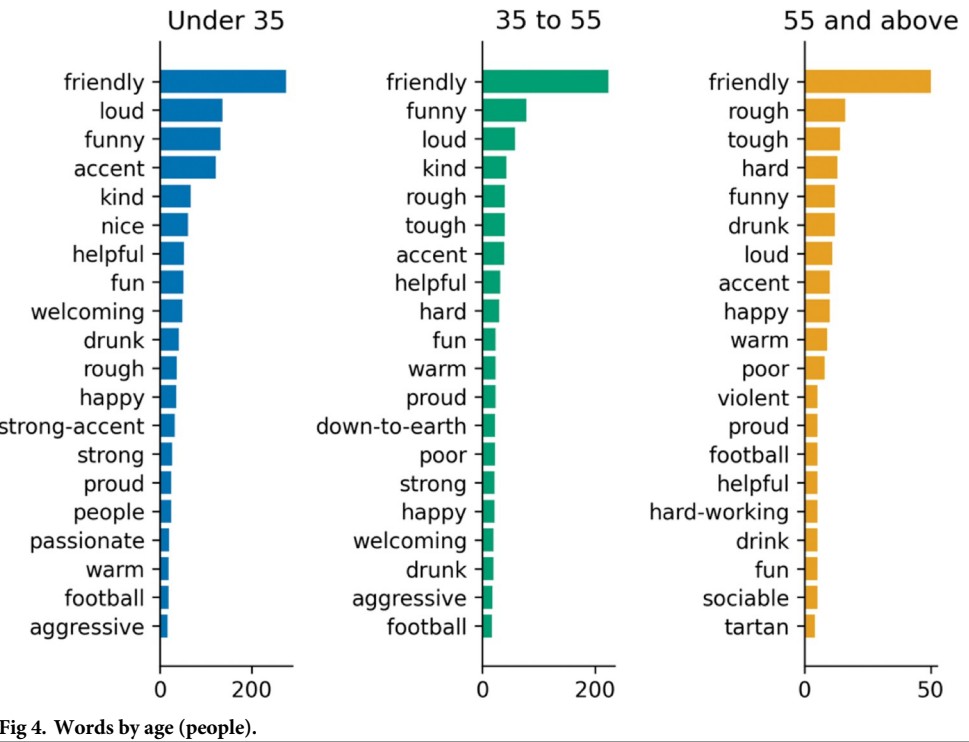

**Fig 4. Words by age (people).**

for younger ages. Clyde (the name of a river that flows through Glasgow) is also mentioned a lot more frequently. Older people also think more often of tenements, a traditional type of Scottish building. All in all, culture seems to be a relevant aspect of Glasgow for older people both through word associations (second most frequently cited), and the "Appeal of Glasgow" question."

Descriptions for the people of Glasgow are quite similar across age ranges, though for 55+, people are seen as rough and tough in addition to being friendly, ahead of funny and loud (second and third for younger respondents).

**4.2.3 Words by gender.** Descriptions of the city vary slightly by gender (Figs 5 and 6), in terms of priorities, *football* is by far the most frequently mentioned term for men, and *cold* the second most frequent, while the top two are reversed for women and much closer in frequency. *Shopping*, *art*, and *buildings* feature among the top 20 for women, though not for men. *Culture* is common to both, and place 3rd for men and 6th for women.

People are viewed in a similar way across gender, being rated as friendly, funny, and loud by both men and women.

**4.2.4 Words by relationship to Glasgow.** It was hypothesized that residents (self) would use a different linguistic style to people who have never visited (other) and hold different views of the city and its people. The following charts (Figs 7 and 8) provide frequencies based on the relationship to Glasgow.

Fig 7 provides properties generated by respondents in ranked order of frequency, by their relationship to Glasgow. Relative frequencies vary across groups with *football* and *cold* topping the list for non-residents, while residents (and former residents) rated *friendly* as the top descriptor. Friendly does not feature at all in the top 20 terms form those that have never been to Glasgow, while for those that have visited or live(d) in the city, *friendly* is rated 6$^{th}$ for visitors and is the top result for residents. All agree that the city is *busy*.

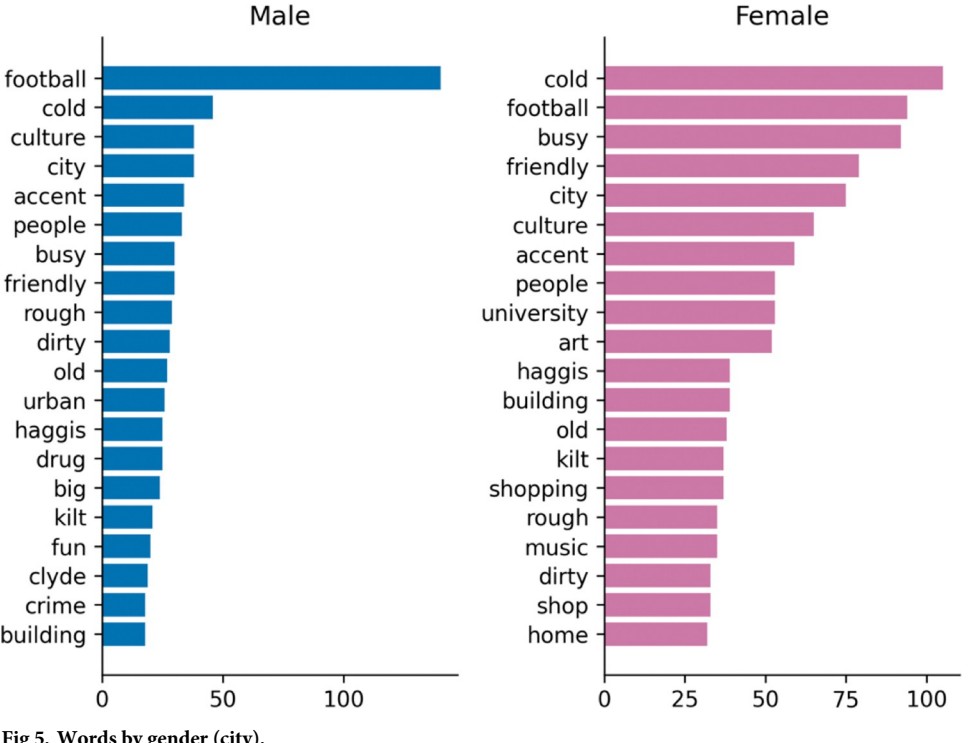

**Fig 5. Words by gender (city).**

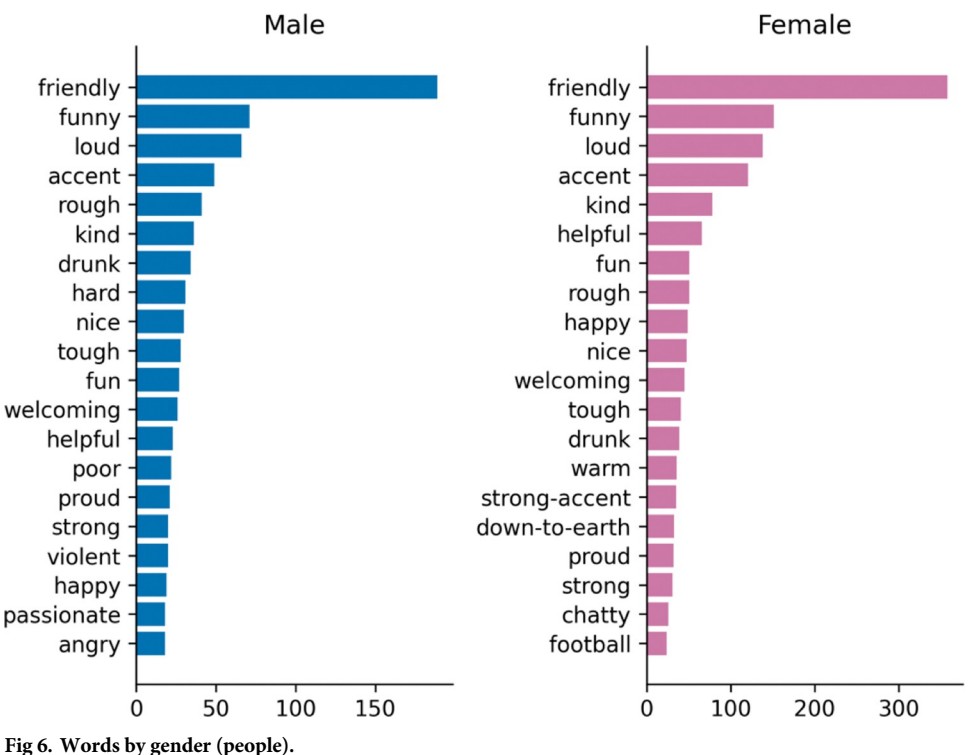

**Fig 6. Words by gender (people).**

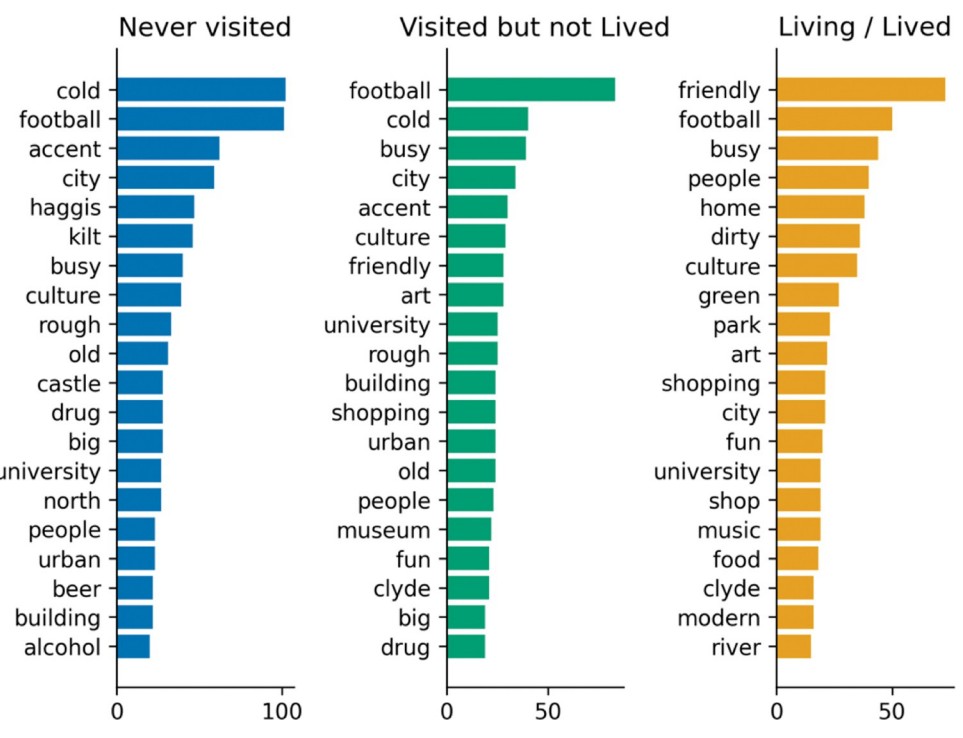

**Fig 7. Words by relationship to Glasgow (city).**

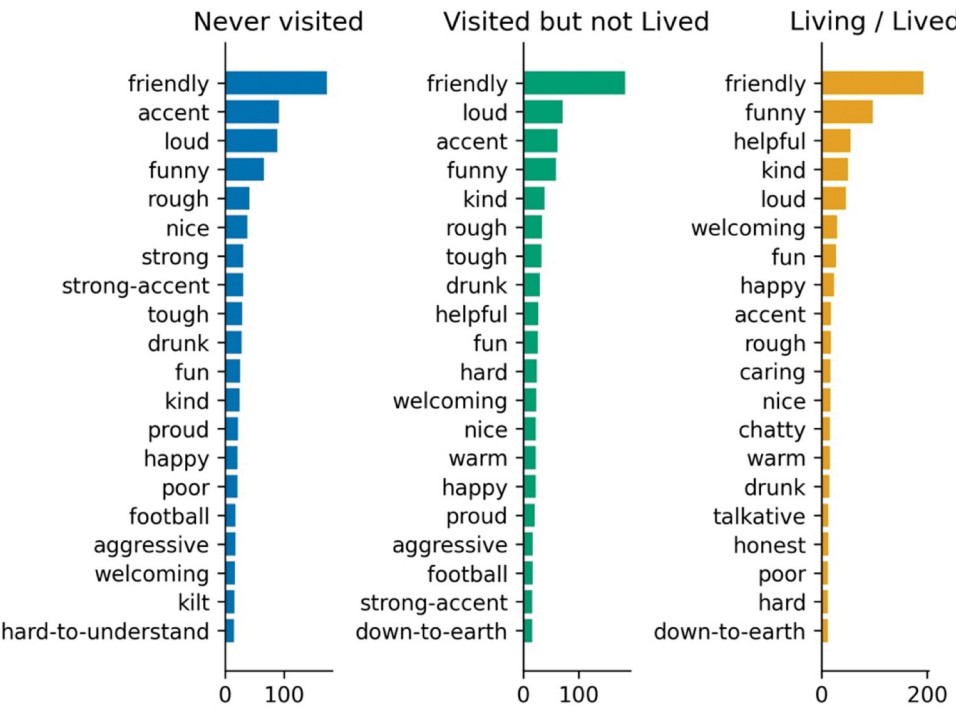

**Fig 8. Words by relationship to Glasgow (people).**

People who live (residents) or have lived in Glasgow use the word friendly a lot more frequently to describe the people of Glasgow compared to people who have never lived there. Football is also lower in the list of terms however still very relevant. All agree that Glaswegians are *funny*, *loud* and *kind*, though Glaswegians do not rank being *loud* as highly as the other groups, who have it at second place.

As described in Section 3.5, word embeddings were calculated and grouped with the adjective that they were closest to in the Aaker dimensions (which included 42 adjectives across the five dimensions).

## 4.3 Resident and visitor preferences (descriptive)

The largest appeal of Glasgow (Fig 9) were the food and hospitality experiences, followed by museums and galleries and built heritage. Sport and meeting facilities were the least frequent appeals. However, differences can be observed by gender, as well as age groups. For men, sport is significantly more important than for women, whereby retail is a substantial appeal for women. Furthermore, there is a steady increase in appeal of museums and galleries as well as built heritage from lower to higher age groups, whereas the appeal of nightlife drops across the lifespan.

## 4.4 Hypothesis tests

**4.4.1 H1: Linguistic and Situated Simulation (LASS).** Paired sample's t-test revealed that the last words were on average more semantically distant from the words Scotland (t = 5.43, p<0.001, d = 0.22), city (t = 5.88, p<0.001, d = 0.22) and town (t = 8.07, p<0.001, d = 0.31) than first words as per H1a. Significant differences were also found for Glasgow (t = 3.88, p<0.001, d = 0.15). With the exception of "Glasgow", which exhibited a slightly smaller effect

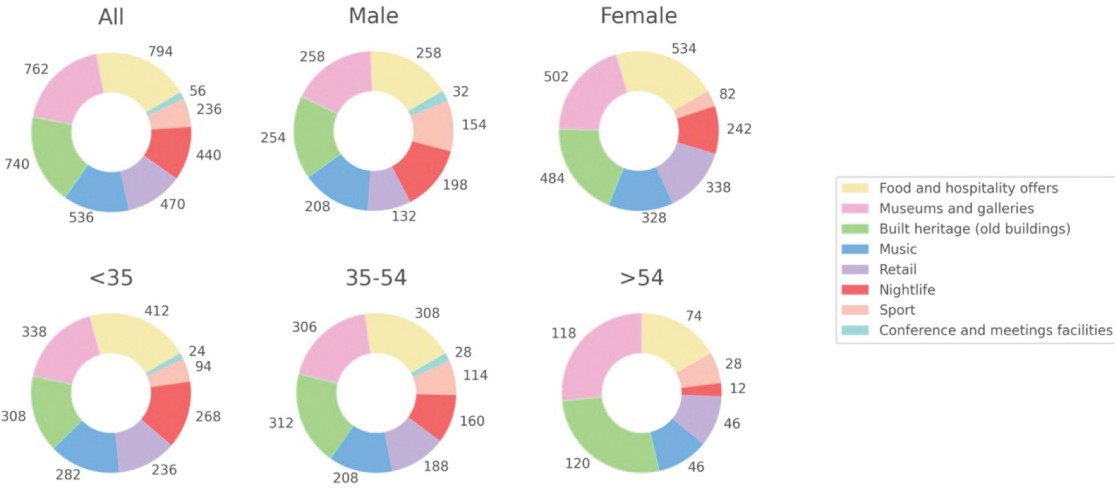

**Fig 9. Appeal of Glasgow.**

size, significant support, with moderate effect sizes could be found for the primary prediction of H1 that properties generated by the linguistic system are semantically closer to the core concept of the cue word than object-situation responses.

For H1b, last words had greater variance and uniqueness than first words, with close to twice as many unique terms (779 unique terms compared to 438), and unique last words were more semantically distant from each other, revealing higher overall variability (t = 3.34, p<0.001, d = 0.21), constant with the hypothesis that last words will more likely be personal, unique, situated simulations.

Overall, support for H1 was found, with last words both semantically more distant from first words, and exhibiting greater internal variance and diversity, as predicted.

**4.4.2. H2: Aaker-dimensions and relationship to Glasgow.** A logistic regression (Table 3) revealed that participants who described Glasgow's city or its people as sincere or exciting had 6.3 times and 1.6 times higher odds to be residents, respectively, 95% CI [3.09, 12.68] and 95% CI [1.09, 2.37]. Inversely, odds for having never visited was 3.29 times higher in people who described Glasgow as rugged and 3.8 times higher for those describing Glasgow as sophisticated, 95% CI [2.14, 5.06] and 95% CI [2.48, 5.65]. Competence did not differ significantly between groups in ether model.

In summary, in line with H2, perception across the Aaker dimensions varied significantly by relationship to Glasgow, particularly Sincerity (positively), Sophistication and Ruggedness (both negative predictors of residence). Finally, word count was a significant predictor, with (ex) residents providing higher numbers of associations than never visitors (odds ratio of 1.54). Overall, the complete model explained 28.7% of the variance in relationship to Glasgow, while the reduced model still predicted 24.5% of the variance.

## 4.5 Exploratory analyses

In addition to the pre-registered hypotheses, new questions emerged as we processed the data that may be interesting and relevant to scholars and practitioners.

**4.5.1 Exploratory factor models for Glasgow city.** After removing all the words with less than 10 occurrences, 101 words for the city question and 79 words for the people question remained. The sample for both analyses was N = 1219. Initially, no factor loading cut-off value was chosen due to an overall low variable to factor correlations, which originated from the low

**Table 3. Predicting the relationship to Glasgow via words coded to the Aaker dimensions.**

| Logistic Regression Results | | |
|---|---|---|
| | **Dependent Variable** | |
| | **Relationship to Glasgow** | |
| | **(1)** | **(2)** |
| **Sincerity (Aaker)** | 1.82*** | 1.83*** |
| | (0.35) | (0.36) |
| **Excitement (Aaker)** | 0.45* | 0.47* |
| | (0.19) | (0.20) |
| **Competence (Aaker)** | 0.34 | 0.31 |
| | (0.19) | (0.19) |
| **Sophistication (Aaker)** | -1.4*** | -1.32*** |
| | (0.20) | (0.21) |
| **Ruggedness (Aaker)** | -1.03*** | -1.19*** |
| | (0.21) | (0.22) |
| **Word Count** | 0.40*** | 0.43*** |
| | (0.05) | (0.05) |
| **Male** | | -0.21 |
| | | (0.20) |
| **Age** | | 0.04*** |
| | | (0.01) |
| **Secondary school (Highest)** | | -0.55 |
| | | (0.32) |
| **College (Highest)** | | -0.59* |
| | | (0.24) |
| **Postgraduate Degree (Highest)** | | 0.51* |
| | | (0.26) |
| **Part time (Employment)** | | 0.33 |
| | | (0.25) |
| **Student (Employment)** | | 0.57 |
| | | (0.31) |
| **Not employed (Employment)** | | -0.26 |
| | | (0.33) |
| **Retired (Employment)** | | -0.32 |
| | | (0.66) |
| **Disabled (Employment)** | | -1.19 |
| | | (0.80) |
| **R-squared** | 0.2451 | 0.2872 |
| **N** | 778 | |

Significance

* p < .05

** p < .01

*** p < .001

number of occurrences for most words. Following processing with higher cut-offs, the cut-off value was finally set to 0.10, as words below this value seemed to be assigned to a factor at random, while words above the threshold were reasonably related to the factor following appraisal and agreement by the authors. Despite the low cut-off values for the factor loadings, these new

**Table 4. Exploratory brand dimensions for Glasgow city.**

| Recreation | Trouble | Excitement | Weather |
|---|---|---|---|
| park | dirty | vibrant | cold |
| museum | poverty | busy | north |
| people | urban | loud | snow |
| center | rough | lively | wet |
| art | busy | friendly | grey |
| shop | poor | fun | northern |
| green | crime | exciting | popular |

categories provide interesting insights for future research and are provided and discussed in that context.

Four factors were identified for city of Glasgow: *recreation*, *trouble*, *excitement*, and *weather* (see Tables 4 and 5). All factors and their relative weights are provided in Table 5, while they are summarised graphically in an association word cloud in Fig 10.

**4.5.2 Exploratory factor models for the people of Glasgow.** For the people of Glasgow, a five-factor approach describe Glaswegians in terms of *warmth*, *ruggedness*, *aggression*, *gregarious* and *substance use* (see Table 6). Factor analysis results are provided in Table 7, while they are graphically summarised in an association word cloud in Fig 11.

## 5. Discussion

### 5.1 On second thoughts (H1)

The order of words may indeed represent a greater or lesser focus on systems dealing with linguistic associations versus situated simulations. It was hypothesized that properties generated by linguistic mechanisms would necessarily be collocated more frequently (be semantically closer) with the cue word than those regarded as situated simulations or object-situation responses, due to the latter being more personal and individual (semantically distant). Following a similar rationale, Santos et al. [46] predicted significantly more unique terms would be generated by object-situation responses than linguistic system-based responses. Indeed, across all words, last words were more distant from all core concepts than first words, and 1.78 times as many unique terms were generated for last terms than first. Supporting the hypothesis that two systems may be active in word association tests, earlier associations appealed to more generally available conceptual models (thereby closer to the core concepts and less diverse in expression). Meanwhile, later associations appeared further from the core concept and were semantically more diverse.

This has an important implication for destination brand research: Rather than aggregating all terms and obtaining an average, looking at word order may help to induce dimensions of a destination that would otherwise be lost. These findings suggest that observing the emergence of situated simulation at scale may be achievable with readily available, cheap, and scalable computational methods. These findings lend themselves to replication studies as they may represent evidence for an important new relationship in the generation of concepts and related properties.

### 5.2 From marketers to people: Brand personality (H2)

The Brand Wheel (Fig 1) distilled Glasgow's brand personality in 2004 into seven adjectives: *Cosmopolitan*, *Humorous*, *Generous*, *Outgoing*, *Genuine*, *Creative* and *Embracing*. As the Brand Wheel does not differentiate those that had never visited the city from local experience,

**Table 5. Exploratory factor model for Glasgow city.**

| Items | Recreation | Trouble | Excitement | Weather |
|---|---|---|---|---|
| park | **0.311** | 0.033 | -0.014 | 0.028 |
| museum | **0.284** | 0.03 | -0.073 | 0.007 |
| people | **0.264** | 0.02 | 0.015 | 0.018 |
| center | **0.233** | 0.019 | -0.067 | 0.059 |
| art | **0.222** | 0.043 | -0.059 | -0.044 |
| shop | **0.218** | 0.021 | 0.036 | 0.012 |
| green | **0.21** | 0.004 | -0.027 | 0.04 |
| food | **0.207** | -0.001 | **0.12** | 0.02 |
| music | **0.197** | 0.022 | 0.032 | -0.066 |
| home | **0.193** | 0.066 | **0.14** | 0.081 |
| culture | **0.17** | 0.03 | -0.044 | -0.051 |
| clyde | **0.163** | 0.023 | -0.063 | -0.013 |
| friendly | **0.163** | 0.078 | **0.253** | 0.084 |
| restaurant | **0.161** | 0.002 | 0.034 | -0.005 |
| friend | **0.143** | 0.005 | -0.023 | 0.025 |
| architecture | **0.141** | 0.024 | 0.006 | -0.012 |
| george-square | **0.139** | 0.019 | -0.065 | 0.013 |
| happy | **0.136** | -0.001 | **0.172** | 0.025 |
| train | **0.133** | -0.062 | 0.07 | -0.064 |
| shopping | **0.122** | 0.011 | 0.068 | -0.012 |
| glasgow | **0.12** | 0.052 | **-0.131** | -0.036 |
| club | **0.117** | 0.009 | 0.006 | -0.064 |
| nightlife | **0.105** | -0.022 | 0.069 | -0.084 |
| dirty | -0.011 | **0.139** | **0.201** | -0.099 |
| poverty | **-0.13** | **0.132** | -0.059 | **-0.202** |
| urban | -0.059 | **0.127** | 0.085 | -0.018 |
| rough | **-0.151** | **0.117** | 0.001 | -0.038 |
| busy | 0.079 | **0.109** | **0.35** | 0.051 |
| poor | **-0.102** | **0.106** | -0.016 | -0.091 |
| crime | **-0.103** | **0.104** | -0.066 | **-0.205** |
| vibrant | 0.001 | 0.077 | **0.372** | -0.026 |
| loud | -0.042 | 0.022 | **0.341** | -0.052 |
| lively | -0.023 | 0.074 | **0.293** | -0.025 |
| fun | 0.056 | 0.066 | **0.241** | -0.046 |
| exciting | -0.032 | 0.072 | **0.235** | -0.014 |
| diverse | -0.008 | 0.044 | **0.19** | 0.043 |
| good | 0.099 | -0.004 | **0.162** | 0.026 |
| modern | 0.043 | 0.06 | **0.143** | 0.005 |
| dangerous | -0.081 | 0.089 | **0.111** | -0.076 |
| cold | **-0.251** | -0.023 | -0.071 | **0.498** |
| north | **-0.176** | -0.03 | -0.065 | **0.254** |
| snow | **-0.135** | -0.038 | -0.056 | **0.25** |
| wet | 0.004 | 0.067 | -0.024 | **0.185** |
| grey | 0.011 | 0.084 | -0.068 | **0.16** |
| northern | -0.052 | 0.024 | -0.04 | **0.15** |
| popular | -0.055 | 0.082 | 0.087 | **0.148** |
| big | -0.043 | 0.063 | 0.045 | **0.13** |

(*Continued*)

**Table 5.** (Continued)

| Items | Recreation | Trouble | Excitement | Weather |
|-------|-----------|---------|-----------|---------|
| castle | -0.047 | **-0.146** | -0.034 | **0.126** |
| tourist | -0.03 | 0.075 | 0.07 | **0.112** |
| rainy | 0 | 0.026 | 0.052 | **0.108** |
| university | 0.081 | 0.045 | -0.049 | **0.105** |
| rain | 0.08 | -0.024 | -0.075 | **0.104** |

or distinguish the city from its people, the current research tested self-versus other, for city and people, across the Aaker brand personality dimensions, and generated new personality dimensions that extended the Aaker framework (Tables 4–7). Furthermore, the research foundation of such attribution of adjectives in this original Brand Wheel was never made available. As noted, significant and noteworthy differences in perception were found across the Aaker dimensions for residents. In support of the Brand Wheel, the current research found Glasgow perceived as *friendly* and *welcoming* (embracing), *funny* and *kind* (humorous and generous),

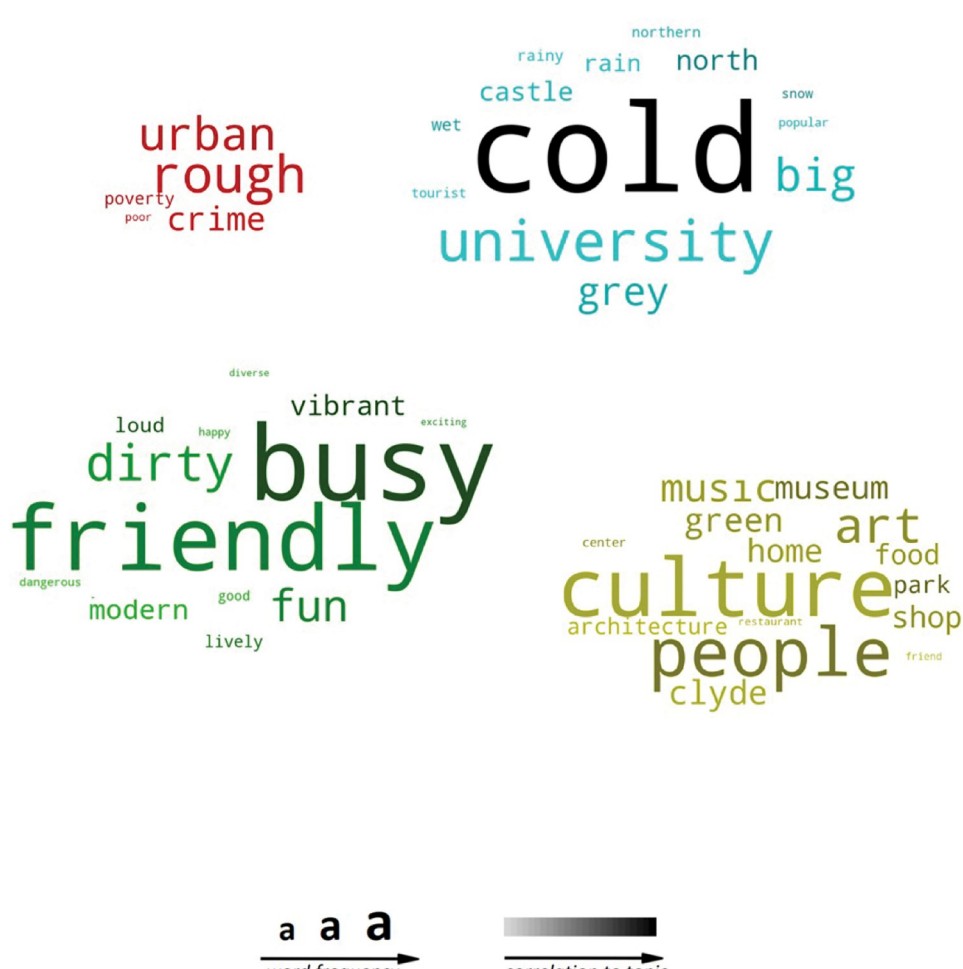

**Fig 10. Word cloud (city), represents the factors from Table 5, arranged by category, word frequency and strength of the correlation (factor loading) within the category.**

**Table 6. Exploratory personality dimensions for the people of Glasgow.**

| Warmth | Ruggedness | Aggression | Gregariousness | Substance Use |
|---|---|---|---|---|
| kind | tough | violent | happy | drug |
| helpful | hardy | common | friendly | alcoholic |
| funny | strong | angry | cheerful | drunk |
| caring | loyal | blunt | chatty | aggressive |
| friendly | hard-working | unfriendly | generous | fight |
| open | hard | poor | talkative | party |
| honest | rough | open | noisy | football |

while the city itself was described as *busy* and *cold.* The range of predominantly positive and complimentary characteristics will rarely raise any objection but when compared with the perceptions and associations offered in this paper it is clear that negative and derogatory associations (aggressive, troublesome, cold) may be a more accurate reflection of urban perceptions.

It is also of interest to note that Brand Wheel positioning is essentially a marketing technique, and the availability of data regarding sampling and validation for the case of Glasgow is absent, making it impossible to interrogate the methods or results. However, incredulity is further extended in the Brand Wheel incorporation of statements and adjectives constructed around sub-sections:

- How the brand makes me Look.

- How the brand makes me Feel.

- What the product does for Me.

Around 25% of the variance in the relationship to Glasgow could be explained by differences in terms within the Aaker dimensions (Section 4.4.2, Table 3). *Sincerity* was a positive predictor of a close relationship to the city, with those describing the people of Glasgow as sincere 6.3 times more likely to be residents. Contrarily, *Sophistication* and *Ruggedness* were negative predictors, with those describing Glasgow as rugged over three times (3.24) less likely to be a (former) resident, and those describing Glasgow as sophisticated almost four times (3.8) less likely to be (former) residents. Finally, residents and former residents were considerably more gregarious in the generation of terms to describe the city. Overall, (former) residents and non—visitors had significantly different views of the city across three of the five Aaker dimensions, indicating a clear opportunity for destination branding and marketing professionals to explore when developing campaigns. This gap can be seen when considering the Brand Wheel.

The contention that there will be a uniformity of perception in such areas simplifies the complexity of human perception and emotional response. The investment in such unvalidated and under theorised work is interesting when contrasted with the limited traction such terms enjoy amongst residents, visitors and non-visitors in the present study. The absence of the proposition: 'Full-on European City Experience' along with almost all the other positively attributed descriptors raises concerns about the rigor of marketing campaigns over logical examination and research. Potentially mitigating some of the challenges noted, the present study also led to the generation of several new trends.

## 5.3 Exploratory analyses

**Football.** Ranked fourth overall across all responses (Fig 2) after *cold* (first), *busy* (second) and *city* (third), it was the most frequently cited association for Glasgow City by age across all groups (Fig 3), top for men and second for women (Fig 5), and even by relationship to

**Table 7. Exploratory factor model for the people of Glasgow.**

| Items | Warmth | Ruggedness | Aggression | Gregariousness | Substance Use |
|---|---|---|---|---|---|
| kind | **0.397** | **-0.104** | **-0.103** | -0.062 | -0.069 |
| helpful | **0.366** | **-0.105** | **-0.185** | 0.045 | -0.03 |
| funny | **0.302** | -0.061 | -0.011 | 0.058 | -0.052 |
| caring | **0.27** | -0.001 | **-0.104** | -0.012 | -0.015 |
| friendly | **0.265** | -0.086 | **-0.208** | **0.248** | -0.067 |
| open | **0.194** | 0.043 | **0.151** | 0.071 | -0.079 |
| honest | **0.19** | 0.004 | 0.075 | -0.017 | -0.059 |
| blunt | **0.138** | -0.029 | **0.198** | 0.039 | -0.067 |
| welcoming | **0.107** | -0.031 | -0.053 | 0.065 | **-0.102** |
| tough | -0.039 | **0.431** | -0.047 | 0.015 | -0.024 |
| hardy | -0.071 | **0.294** | -0.077 | 0.018 | -0.046 |
| strong | 0.017 | **0.28** | -0.027 | -0.049 | 0.015 |
| loyal | 0.055 | **0.209** | 0.016 | -0.039 | -0.056 |
| hard-working | -0.003 | **0.197** | -0.058 | 0.008 | -0.084 |
| hard | 0.081 | **0.164** | 0.089 | 0.017 | 0.092 |
| rough | -0.03 | **0.158** | **0.111** | 0.027 | 0.035 |
| working-class | -0.025 | **0.137** | 0.082 | 0.012 | -0.056 |
| violent | -0.033 | -0.025 | **0.34** | 0.083 | 0.072 |
| common | 0.022 | 0.034 | **0.255** | 0.022 | -0.098 |
| angry | -0.032 | 0.009 | **0.236** | 0.046 | 0.081 |
| unfriendly | -0.041 | -0.034 | **0.171** | 0.011 | 0.001 |
| poor | 0.005 | 0.017 | **0.154** | 0.055 | **0.149** |
| rude | -0.069 | -0.036 | **0.136** | 0.008 | 0.084 |
| brash | -0.046 | 0.003 | **0.127** | 0.044 | -0.047 |
| drunk | -0.036 | -0.074 | **0.113** | 0.09 | **0.233** |
| interesting | 0.046 | -0.016 | **0.106** | -0.021 | -0.083 |
| happy | **-0.117** | **-0.208** | **-0.192** | **0.296** | -0.038 |
| cheerful | **-0.143** | **-0.121** | **-0.135** | **0.191** | -0.037 |
| chatty | 0.044 | -0.078 | 0.003 | **0.177** | -0.041 |
| generous | 0.083 | -0.021 | -0.042 | **0.151** | -0.018 |
| talkative | 0.025 | -0.053 | 0.086 | **0.142** | -0.02 |
| noisy | -0.043 | -0.048 | 0.085 | **0.135** | -0.044 |
| banter | 0.043 | -0.057 | -0.084 | **0.122** | 0.055 |
| down-to-earth | 0.029 | 0.002 | 0.02 | **0.117** | -0.044 |
| sociable | -0.01 | 0.007 | -0.036 | **0.116** | -0.019 |
| fun | 0.041 | -0.043 | **-0.15** | **0.115** | -0.024 |
| approachable | 0.079 | -0.077 | **-0.134** | **0.11** | **0.1** |
| drug | -0.025 | -0.021 | 0.016 | -0.029 | **0.418** |
| alcoholic | 0.002 | 0.023 | 0.034 | -0.022 | **0.347** |
| aggressive | -0.047 | 0.061 | 0.064 | 0.016 | **0.209** |
| fight | -0.021 | 0.08 | -0.019 | -0.012 | **0.194** |
| party | -0.021 | -0.05 | -0.058 | -0.031 | **0.179** |
| football | -0.029 | 0.018 | -0.045 | 0.011 | **0.162** |

Glasgow (Fig 7), rated second to *cold* for non-visitors, top for visitors, and second for residents (second to *friendly*). The presence of Football as the most popular sport in the world for the last century which has been further propelled through media and digital channels (YouGov

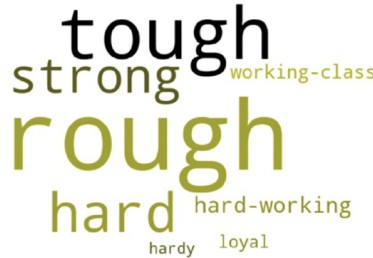

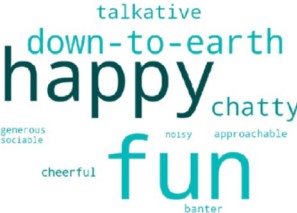

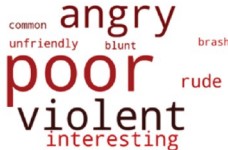

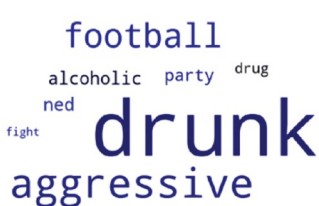

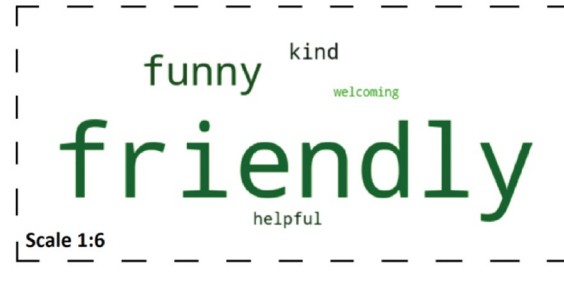

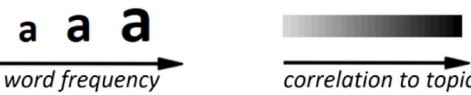

**Fig 11. Word cloud (people), represents the factors from Tables 6 and 7, arranged by category, word frequency and strength of the correlation.**

Sport, 2022, https://sport.yougov.com/global-just-how-popular-is-football/), should not be a surprise. The absence in the destination marketing literature of this common word association between football and Glasgow further raises questions about the distance between marketing hyperbole and the perception of consumers' on the city of Glasgow.

**Cold.** Glasgow was also perceived as *cold*, across all demographics, with the strongest overall association for the city in the factor model (.584, Table 5). In fact, Glasgow's weather appeared so prominently that it managed to distinguish itself as an entire factor in the exploratory model. Although the top-rated term describing the city for non-visitors, and second for visitors, the *cold* did not make it to the top 20 terms for (former) residents (Fig 7). What do residents see that visitors and non-visitors do not? At the top of the list, they think of Glasgow in terms of *people*, with *friendly* their most frequently cited term for the city.

**Poverty and trouble.** Table 4 and 5 (Fig 10) also indicate a category relating to trouble and poverty, while substance use featured in perception towards Glasgow's people in Tables 6

and 7. Perhaps most telling, the association and frequency cloud for the People of Glasgow (Fig 11) highlights that prevalent negative associations remain. Despite the positive association with friendliness, Glasgow's history of substance misuse and violence remains. The differences in perceptions of city and people are also marked enough to warrant any city branding to consider both dimensions. Prior studies of Glasgow city appear to have largely ignored an anthropocentric approach to destination marketing, despite '*People make...*' being the city's current strapline. What does this mean for Glasgow city, if *people make it*?

**People make Glasgow.**   Response types varied when asked to consider the city versus its people, and the extended dimensions derived from modelling the terms used to describe Glasgow city and people provided two different models for each. For example, *excitement* and *recreation* (similar to Aaker's *sophistication*) can be found in perceptions of the city, whereas *warm* and *gregarious* (similar to *sincerity*) and *hardy*, *aggressive* and references to *substance use* (possible elements of *Ruggedness*) are found in perceptions of the people of Glasgow. A review of the words generated led to the development of two exploratory factor models for the city (Tables 4 and 5, Fig 10) and people of Glasgow (Tables 6 and 7, Fig 11). The words used to describe Glasgow, when modelled separately, provide a portrait of Glasgow that extends the perception of the city in terms of *recreation*, *trouble*, *excitement*, and *weather*, while its people are *warm*, *rugged*, *aggressive*, and *gregarious*, and invoke associations with *substance use*.

Despite marketing Glasgow in respect of people, through *People Make Glasgow*, when asked to think of the city and the people separately, only (former) residents extended the top cited term for people to their top association of the city (Fig 7). *Friendly* made it to sixth place for visitors, while it did not make the top 20 for non-visitors, suggesting the perception of Glasgow as a friendly city, may not have travelled conceptually. Despite this, Glasgow's people (Fig 8) were seen as friendly by all groups, so the absence of this connection to the city for non-visitors may suggest a failure of the people-centric marketing of *People Make Glasgow* to reach those unfamiliar with the city, or to make the connection that Glasgow is, largely, its people. This finding represents an opportunity for marketers to help others see Glasgow, as Glaswegians see it, considering Glasgow's people are already perceived as friendly and warm, seeing these terms rise in associations with the city itself would represent a viable measure of success for the campaign, while any associations of warmth, may help negate the frequency with which those less familiar with the city regard Glasgow as *cold*.

The fact that aggression, violence and drug and alcohol use were cited frequently enough to form their own factors, is also a motivating factor for future research. Indeed, the ruggedness ascribed to Glasgow, more prominently by outsiders than residents, may be impacting Glasgow's appeal. The perceptions of locals could make stronger connections with positive imagery and experiences.

The results suggest using both the city AND people question when conducting destination branding studies of this nature. Although people may make Glasgow, they are not inseparable from it in the minds of non-visitors, which may impact the overall perception of Glasgow as a preferred destination. Furthermore, such analysis challenges government agency expenditure on largely unvalidated and non-consumer tested campaigns. Forensic analysis of language and association will only help destination marketers produce more successful campaigns.

## 6. Limitations

The quest for representativeness remains a challenge regardless of the sampling strategy used with each having their own limitations such as response rates, online or data literacy and so forth [71] and the present study is not exempt from such limitations. The present study was conducted using an online panel service, and as such, the findings may not generalize to

another, or indeed, the entire UK population. Additionally, as the survey was voluntary, self-selection biases may limit the representativeness of the sample. By providing the data and codebook the authors hope to facilitate validation, alternate approaches to segmentation and general reuse.

## 7. Conclusion

The study contributes to theory and practice in several meaningful ways. In terms of theory, the paper proposed analysing word order in terms of semantic distance, as a means of identifying the extent to which respondents may have invoked memories (situated simulations) over common language associations in word association tasks (H1). Evidence was found in support of the idea that both greater average semantic distance from core concepts and greater uniqueness in terms generated may provide a workable proxy for identifying which systems of cognition may be active in a word association task (i.e. in terms of Dual Code Theory, language associations versus situated simulation) over costly approaches such as fMRI scans, representing a contribution to LASS theory, and grounded cognition. To date, the present study appears to be unique in considering the role of dual code theory on word associations applied to destination marketing. One practical implication of this finding (to be tested further in future studies), is that when looking at word association studies, the generation order of terms may indicate two levels of association: an immediate, more generic, and stable image of a city (through common language associations) and a spectrum of situated associations, offering opportunities for a more personalised approach to destination marketing. In short, the present study represents an initial foray into exploring the potential implications of dual code theory on the generation of word associations in the context of destination branding, and that differences may exist in personal associations versus culturally embedded interpretations of the city concept.

With regards to Aaker's theory of brand personality, the study contributes to nascent research into the multiple personalities a destination may possess, when considered from the perspective of an observer's relationship to the city and offers an opportunity to extend existing theory through recognition that relationships to a destination may substantially impact perception (H2). The study found evidence for *a tale of two cities*, and while the differences found were not as revolutionary as those in the Charles Dickens' novel, residents appear to experience a subtly different Glasgow to that of visitors, or imagined by non-visitors. In subsequent exploratory analyses, possible extensions to the Aaker dimensions were also identified for Glasgow. The extent to which this may be true for other destinations, and particularly urban destinations, offers a rich opportunity to extend existing theory but also influence government agencies to target more effective destination marketing techniques. It is apposite that at a time of constrained resources in the UK the appropriateness of marketing investment that provides clear perceptual impacts can be scrutinised and judged.

Practically, while the failure of marketing professionals to appreciate the dichotomy is illustrative of the untested nature of such investment, it is also an opportunity. With regards to Glasgow and its people, the present findings question the orthodoxy of a city that has supposedly reinvented itself and moved on from violence, drugs and negative perceptions, while the simple dominance of football association, rather than any of the promoted marketing terms is also indicative of the superficial penetration of investment in destination marketing campaigns since 1983. Indeed, variations in perception from non-visitors to visitors or residents, illustrates a need to target these groups according to their familiarity with the city, and the presented methods provide marketers with tools to quickly obtain a snapshot into targeted perceptions.

Traditional approaches to tourism marketing and destination branding rarely provide such a level of analytical appraisal. The methods presented in this study offer destination marketers and researchers a simple, replicable approach to obtaining qualitative perception data in a controlled manner, at scale. Furthermore, the use of NLP detailed provides a fast and cost-effective way of matching associations to either predefined categories such as the Aaker dimensions, or any other terms of interest. In the case of Glasgow, these methods could be used to measure the effectiveness of campaigns such as *Scotland with Style*, by measuring the semantic distance of terms generated by respondents to the word *style*, or they could be used to identify potential terms for future marketing campaigns.

Despite the short time investment required to gather data for this study, the richness of insights and interpretation that could be derived were, arguably, disproportionately high with evidence for the perception drawn from both immediate associations and those derived from lived experience of the city. Automated destination personality assessment represents a promising opportunity that has only begun to be explored. Prior studies of Glasgow city appear to have largely ignored or omitted an anthropocentric approach to destination marketing. This study represents an important and reproducible step away from marketing hyperbole towards greater nuance in destination marketing and consumer perception studies, and for Glasgow, many new insights and opportunities to leverage the authenticity and warmth of its people.

## Supporting information

**S1 Appendix. In other words: A short word association game.**
(DOCX)

## Author Contributions

**Conceptualization:** Sean MacNiven.

**Data curation:** Maxime MacNiven.

**Formal analysis:** Maxime MacNiven.

**Funding acquisition:** J. John Lennon.

**Investigation:** Sean MacNiven.

**Methodology:** Sean MacNiven, Maxime MacNiven.

**Project administration:** Sean MacNiven.

**Software:** Maxime MacNiven.

**Supervision:** Sean MacNiven.

**Writing – original draft:** Sean MacNiven, J. John Lennon, Julie Roberts, Maxime MacNiven.

**Writing – review & editing:** Sean MacNiven, J. John Lennon, Julie Roberts, Maxime MacNiven.

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
