## [Decision Letter · Decision Letter 0]

7 Jul 2023

PONE-D-23-11026

In other words: Glasgow (word association study)

PLOS ONE

Dear Dr. MacNiven,

Thank you for submitting your manuscript to PLOS ONE. After careful consideration, we feel that it has merit but does not fully meet PLOS ONE’s publication criteria as it currently stands. Therefore, we invite you to submit a revised version of the manuscript that addresses the points raised during the review process.

I agree with the Reviewer’s comments regarding the method description and your conclusions. On top of that, please improve the clarity of your text, explain your aim (and the corresponding research gap and your research question(s); they should also motivate your exploratory analyses) in the introductory part, and show the theoretical and practical importance of your results in the conclusions. Please find my detailed comments in the attachment.

We look forward to receiving your revised manuscript.

Kind regards,

Wojciech Trzebiński, Ph.D.

Academic Editor

PLOS ONE

Journal Requirements: 

Reviewers' comments:

Reviewer's Responses to Questions

**Comments to the Author**

1. Is the manuscript technically sound, and do the data support the conclusions?

Reviewer #1: Yes

2. Has the statistical analysis been performed appropriately and rigorously? 

Reviewer #1: Yes

3. Have the authors made all data underlying the findings in their manuscript fully available?

Reviewer #1: Yes

4. Is the manuscript presented in an intelligible fashion and written in standard English?

Reviewer #1: Yes

5. Review Comments to the Author

Reviewer #1: Thank you for the invitation. The research looks decent yet following issues should be addressed for further consideration

1.The full details of word association procedures should be presented.

2.The authors must be clear of the sample characteristic. For example, among 1219 participants, how many are residing residents, first-time tourists and repeated tourists. Based on the current report, my understanding is 277 residing residents, 418 visited tourists, and 524 never been to Glasgow. Please do correct me if I am wrong.

3.3.2 survey is referring to the instrument that should be placed before 3.1 participants.

4.3.3 Natural Language Processing is based who’s suggestion?

5.The authors must address how does this study value add to the existing tourism literature.

6.Also, how could these findings offer insights for practitioner in managing Glasgow destination image. This must be addressed to reflect the usefulness of the results on to the real-life situation.

6. PLOS authors have the option to publish the peer review history of their article (what does this mean?). If published, this will include your full peer review and any attached files.

Reviewer #1: No

---

## [Author Response · Author response to Decision Letter 0]

14 Aug 2023

Dear Reviewer and Editor, thank you for your time and thoughtful commentary. We believe the comments made were valuable and have attempted to address all comments made to the best of our ability. We believe the paper is much improved thanks to your comments, however, leave further deliberation, and hopefully a positive decision naturally with you. We are happy to make any further changes deemed necessary and await your decision. A detailed response and notes about where changes have been made is available in the dedicated document attached. Many thanks again. Sean MacNiven and the team.

---

## [Decision Letter · Decision Letter 1]

25 Aug 2023

PONE-D-23-11026R1In other words: Glasgow (word association study)PLOS ONE

Dear Dr. MacNiven,

Thank you for submitting your manuscript to PLOS ONE. After careful consideration, we feel that it has merit but does not fully meet PLOS ONE’s publication criteria as it currently stands. Therefore, we invite you to submit a revised version of the manuscript that addresses the points raised during the review process.

Thank you for improving your manuscript. However, some major concerns remain. Reviewer 1 provided several comments on your analysis. On top of that, here are my comments: (1) The clean version of the manuscript differs from the mark-up version (e.g., look at the Abstract; the last sentence in the mark-up version is absent in the clean version). Therefore, I cannot evaluate the communication of your revised manuscript. Please fix it. (2) Exploratory analysis: It looks like this analysis aims to answer some additional research questions (you mentioned that in row 659 of the mark-up version). Please explicitly state those questions before presenting that analysis. Were your hypotheses preregistered? If so, please specify the preregistration platform. If not, please do not call them so (e.g., row 659 of the mark-up version).

We look forward to receiving your revised manuscript.

Kind regards,

Wojciech Trzebiński, Ph.D.

Academic Editor

PLOS ONE

Reviewers' comments:

Reviewer's Responses to Questions

**Comments to the Author**

1. If the authors have adequately addressed your comments raised in a previous round of review and you feel that this manuscript is now acceptable for publication, you may indicate that here to bypass the “Comments to the Author” section, enter your conflict of interest statement in the “Confidential to Editor” section, and submit your "Accept" recommendation.

Reviewer #1: All comments have been addressed

2. Is the manuscript technically sound, and do the data support the conclusions?

Reviewer #1: Partly

3. Has the statistical analysis been performed appropriately and rigorously? 

Reviewer #1: Yes

4. Have the authors made all data underlying the findings in their manuscript fully available?

Reviewer #1: Yes

5. Is the manuscript presented in an intelligible fashion and written in standard English?

Reviewer #1: Yes

6. Review Comments to the Author

Reviewer #1: Thank you for the revision while I apperciate the authors' efforts, the results of implicit association require further explanation. The calculation and grouoing of implicit association results is missing. For instance, Harvard Implicit does indicate the categories for each implicit association. Based on reaction time and further calculation, each respondents can be classified into one of the groups. Hence, what is corresponding calculation and grouping for this research. For example, -0.15 to 0.15 is noted as neutral; 0.15 -0.349 is slightly positive; 0.35 - 0.649 is moderate while o.65 to 2.0 is strongly. Vice versa for the negative values. So, what is ditribution of such categorization？

Next, for the sample, about half of them have never been to Glasgow, and their perceptions do exert impacts on the overall results. How does the author account for this interference during analysis?

7. PLOS authors have the option to publish the peer review history of their article (what does this mean?). If published, this will include your full peer review and any attached files.

Reviewer #1: No

---

## [Author Response · Author response to Decision Letter 1]

27 Aug 2023

Dear Reviewer. Thank you for your time. You made a comment about the use of the IAT as a method. This study does not employ that method (we employee a timed, free-word association method) and so no changes have been made with regards to the use of an IAT method.

---

## [Decision Letter · Decision Letter 2]

15 Sep 2023

PONE-D-23-11026R2In other words: Glasgow (word association study)PLOS ONE

Dear Dr. MacNiven,

Thank you for submitting your manuscript to PLOS ONE. After careful consideration, we feel that it has merit but does not fully meet PLOS ONE’s publication criteria as it currently stands. Therefore, we invite you to submit a revised version of the manuscript that addresses the points raised during the review process.

We look forward to receiving your revised manuscript.

Kind regards,

Wojciech Trzebiński, Ph.D.

Academic Editor

PLOS ONE

Journal Requirements:

**Additional Editor Comments:**

Thank you for your response to the concern of Reviewer 1. However, there are still multiple issues with the communication in your manuscript that should be improved before your submission can be accepted. Please see my detailed comments in the attachment.

Reviewers' comments:

Reviewer's Responses to Questions

**Comments to the Author**

1. If the authors have adequately addressed your comments raised in a previous round of review and you feel that this manuscript is now acceptable for publication, you may indicate that here to bypass the “Comments to the Author” section, enter your conflict of interest statement in the “Confidential to Editor” section, and submit your "Accept" recommendation.

Reviewer #1: All comments have been addressed

2. Is the manuscript technically sound, and do the data support the conclusions?

Reviewer #1: Yes

3. Has the statistical analysis been performed appropriately and rigorously? 

Reviewer #1: Yes

4. Have the authors made all data underlying the findings in their manuscript fully available?

Reviewer #1: Yes

5. Is the manuscript presented in an intelligible fashion and written in standard English?

Reviewer #1: Yes

6. Review Comments to the Author

Reviewer #1: (No Response)

7. PLOS authors have the option to publish the peer review history of their article (what does this mean?). If published, this will include your full peer review and any attached files.

Reviewer #1: No

---

## [Author Response · Author response to Decision Letter 2]

19 Oct 2023

Dear Editor, 

The changes requested have been made. The manuscript has been proofread and copyedited. Several points requiring clarification have been addressed. All comments provided have been considered and amended to the best of our ability. 

On a side note, many thanks for your time, patience and effort, we are impressed by the level of dedication at PLOS One, and though this has required several iterations, we believe the manuscript has profited from the thoughtful comments provided. We hope with these latest changes that the manuscript now meets PLOS One requirements. If not, however, we will of course continue to amend as necessary. 

Dr. Sean MacNiven

Specific responses (also in attached Word Document).

PLOS One: There are many imprecise or unclear expressions. I believe most of them are caused by language mistakes (e.g., in the sentence “The extent to which word associations varied by participants’ relationship to Glasgow was identified, in terms of Aaker’s brand personality scale, an extension of personality research to brands and destinations.”, why did you put the comma before “interns”?). Another example: you misspelled the word “collocation” (row 61). You write “Covid” instead of “COVID-19”. Your manuscript should be rigorously proofread to be accepted. Those problems create the impression that your manuscript is not digested and also lead to serious problems with understanding your key statements.

Team: The manuscript has been proofread and copyedited for clarity. We hope there are no further issues, however, please flag any should you find them and we will correct (see the track changes version of the manuscript for all changes made). 

PLOS One: For example, in H1, you state that “Word associations written later by the respondent in the timed response are more likely to be situated simulations (SS) than those earlier language associations (LA)…”. Why did you put “those earlier” in that statement? If I understand your idea correctly, it should be,

“Word associations written later by the respondent in the timed response are more likely to be situated simulations (SS) than language associations (LA).”

Next, you write, “SS is operationalised in terms of semantic distance, with SS predicted to be more distant from one another (variance across terms generated) and the core concept of city (extended to tests of Scotland and Glasgow as sub-categories) than LA”. This sentence is completely unclear. What does it mean “from one another”? Again, if I understand

your idea correctly, it should be something like,

“Word associations written later are predicted to be more distant from each another.” 

Maybe discussing that operationalization in the method section would be clearer.

Team: This has been rewritten for clarity. 

In 2.6 Hypotheses: 

“H1: Word associations written later are predicted to be more distant from the city and each other.”

AND

“Word associations generated later in the word association task by the by the respondent are more likely to be situated simulations (SS) than language associations (LA).” 

H1b also rewritten as “Word associations written later are predicted to be more distant from each another”

“Specifically, SS are predicted to be more distant from the core concept of city (extended to tests of Scotland and Glasgow as sub-categories) than LA, and to have higher variability (i.e., terms generated later will be more unique from one another in terms of average semantic distances than the average distances calculated for LA).”

Also added the new working to the method section (3.5). 

“H1b stated that “Word associations written later are predicted to be more distant from each another”. To test this hypothesis, the variance of associations was calculated within the group of all unique first, and all unique last words, respectively.”

PLOS One: It is unclear what the expression “Aaker dimensions are sufficient” means (row 321). It is also unclear what variables were involved in your Exploratory Factor Analysis. It is not explained in the Method section. It remains unclear in the Results section. E.g., how is the variable corresponding to the item “park” defined at the respondent level? (I suppose that respondents are units of your analysis).

You state that your exploratory analyses aim to address the research questions, “What other terms emerge in descriptions of Glasgow and its people, and how might these be used to extend of enhance the Aaker model?”. However, in row 321, you declare another aim of your exploratory analyses (“To see if the Aaker personality dimensions were sufficient.). This is inconsistent. By the way, the expression “other terms” (row 343) is unclear (other vs. what?).

Team: Now changed in Section 3.7 

“To see if the Aaker personality dimensions were utilised by the sample in their timed responses, an exploratory factor analysis with varimax rotation was run separately for both city and people as a method for investigating the co-occurrence of words. This was achieved by creating a binary column for each term of the word association task (for example, “park”), to indicate whether a participant had, or had not used the term. Then, to ensure that the words used had sufficient weight and relevance, words with fewer than ten occurrences were removed. Words that tend to appear together in the same response, were placed into a common category.

Removed this inconsistency. Now aligned with: 

“In addition to the pre-registered hypotheses, additional ad hoc analyses were conducted to explore what other terms outside of the Aaker dimensions may have emerged in descriptions of Glasgow and its people, and how might these be used to extend the Aaker model.”

Also Section 7: Conclusions a sentence added: “In subsequent exploratory analyses, possible extensions to the Aaker dimensions were also identified for Glasgow.”

PLOS One: In the Results section, when you report on differences, please support them with statistical tests (e.g., rows 346-350). Instead of “p = 0.000,” it should be “p < 0.001.”

Team: This has been corrected. 

PLOS One: The sentence “It is the absence of this powerful association in all of the destination marketing examined that is perhaps most telling and reaffirms the distance between marketing hyperbole and perception” is completely unclear. E.g., what do you mean by “powerful association” and destination marketing examined”?

Team: This statement was indeed sweeping and unclear, it has been reworded (Section 5.3): 

“The absence in the destination marketing literature of this common word association between football and Glasgow further raises questions about the distance between marketing hyperbole and the perception of consumers’ on the city of Glasgow”.

PLOS One: Your Discussion section is dominated by exploratory findings, which seem disconnected from the main aim of your study (see, e.g., your Abstract). In general, my impression is that your study aims to investigate two issues related to destination image: (1) LA vs. SS associations in time and (2) the differentiation of destination personality between destination inhabitants and non-inhabitants. If so,two things should be improved. 

First, the above aims should be reflected in the title, which can refer to Glasgow as an example (case) you use.

Second, the readers would like to see the research gap: how your findings advance the literature. Were those two issues studied before in the destination context? You do not explain that. For example, in the theoretical background section, you only say, “The present study employs LASS as a theoretical lens to both categorise and interpret the properties generated and provide an additional opportunity to test predictions made by the theory.”

Conclusions section, you say that “the paper proposed analysing word order in terms of semantic distance, as a means of identifying the activation of different systems in word association tasks.”

Does it mean that some previous studies on tourist destinations already evidenced the LA vs. SS associations in time, and your contribution is only the new method?

Team: This is an excellent point. Section 5.2 has been expanded and renamed “5.2 From Marketers to People: Brand Personality (H2)”. Similarly, Section 5.3 has also been renamed “5.3 Exploratory Analyses”. Although this section has not been shortened, all subsections have been collapsed into paragraphs. The discussion section now comprises three sections 5.1 for H1, 5.2 for H2 and 5.3 for the exploratory hypotheses. 

The title has been changed to: “The Language of Marketing Hyperbole and Consumer Perception – the Case of Glasgow”.

Section 2.3, added: “The present study employs LASS as a theoretical lens to both categorise and interpret the properties generated and provide an additional opportunity to test predictions made by the theory. To date, no study of destination marketing could be found exploring the role of generation order of terms in word associations studies, nor differences in personal associations versus culturally embedded interpretations of the city concept.”

No, this aspect had not previously been explored. We have now made that research gap clearer, and followed on from Section 2.3, in Section 7 with: 

“To date, the present study appears to be unique in considering the role of dual code theory on word associations applied to destination marketing. One practical implication of this finding (to be tested further in future studies), is that when looking at word association studies, the generation order of terms may indicate two levels of association: an immediate, more generic, and stable image of a city (through common language associations) and a spectrum of situated associations, offering opportunities for a more personalised approach to destination marketing. In short, the present study represents an initial foray into exploring the potential implications of dual code theory on the generation of word associations in the context of destination branding, and that differences may exist in personal associations versus culturally embedded interpretations of the city concept.”

---

## [Editor Report · Decision Letter 3]

16 Nov 2023

The Language of Marketing Hyperbole and Consumer Perception – the Case of Glasgow

PONE-D-23-11026R3

Dear Dr. MacNiven,

We’re pleased to inform you that your manuscript has been judged scientifically suitable for publication and will be formally accepted for publication once it meets all outstanding technical requirements.

Kind regards,

Wojciech Trzebiński, Ph.D.

Academic Editor

PLOS ONE
---

## [Editor Report · Acceptance letter]

20 Nov 2023

PONE-D-23-11026R3 

The Language of Marketing Hyperbole and Consumer Perception – the Case of Glasgow 

Dear Dr. MacNiven:

I'm pleased to inform you that your manuscript has been deemed suitable for publication in PLOS ONE. Congratulations! Your manuscript is now with our production department. 

Kind regards, 

on behalf of

Dr. Wojciech Trzebiński 

Academic Editor

PLOS ONE